# The Contribution of Thai Fisheries to Sustainable Seafood Consumption: National Trends and Future Projections

**DOI:** 10.3390/foods10040880

**Published:** 2021-04-17

**Authors:** Tiptiwa Sampantamit, Long Ho, Carl Lachat, Giles Hanley-Cook, Peter Goethals

**Affiliations:** 1Department of Animal Sciences and Aquatic Ecology, Faculty of Bioscience Engineering, Ghent University, 9000 Ghent, Belgium; Tiptiwa@tsu.ac.th (T.S.); Peter.Goethals@UGent.be (P.G.); 2Department of Biological and Environmental Sciences, Faculty of Science, Thaksin University, Patthalung 93110, Thailand; 3Department of Food Technology, Safety and Health, Ghent University, 9000 Ghent, Belgium; Carl.Lachat@UGent.be (C.L.); Giles.HanleyCook@UGent.be (G.H.-C.)

**Keywords:** fishery products, seafood, nutrition, food supply, consumption trend, Thailand

## Abstract

Sustainably feeding a growing human population is one of the greatest food system challenges of the 21st century. Seafood plays a vital role in supporting human wellbeing, by providing bioavailable and nutrient-dense animal-source food. In Thailand, seafood demand is increasing, and wild capture fishery yields have plateaued, due to oceanic ecosystem degradation and fishery stock exploitation. In this study, we investigated the supply trend of fishery products and subsequent seafood-derived nutrient availability over the last decade. In addition, we explored the possibility of predicting seafood availability and consumption levels, including adherence to Thailand’s national food guide and global dietary recommendations for sustainable seafood consumption. Our findings indicate that, at national-level, fishery products supplied between 19% and 35% of the Thai populations recommended dietary protein intake, 4–6% of calcium, 6–11% of iron, and 2–4% of zinc from 1995 to 2015. Nevertheless, our research also reports that if Thailand’s wild-caught seafood production were to decrease by 13%, as is highly likely, by 2030, the country might face a per capita supply deficit of fish and shellfish to meet healthy and sustainable dietary recommendations (28–30 g/day), let alone the current Thai average intake (32 g/day). Although a 1% per year increase in aquaculture production might bridge this supply gap, policymakers and relevant fishery stakeholders must consider the long-term environmental impacts of such an approach in Thailand.

## 1. Introduction

In 2050, our planet is expected to accommodate 9.7 billion people, putting our food systems under substantial pressure to sustainably feed a growing human population [1,2,3]. It is estimated that we will require about 70% more food available for human consumption as compared to present production [4]. Another major challenge is to ensure adequate dietary nutrient intakes [5,6], as at present more than 800 million people globally are incapable of fulfilling their nutritional requirements [7].

Fishery products play a crucial role in supporting human well-being, mainly by providing bioavailable and nutrient-dense food [8,9]. They offer a vital source of animal protein and essential micronutrients, especially for vulnerable populations in low- and middle-income countries [10,11]. Previous studies have investigated the potential of fish species to combat micronutrient deficiencies in various countries [10,12,13]. For example, small fish species have high levels of vitamin A, iron, and zinc [6,12,14]. Thilsted (2012) [13] reported that certain small indigenous fish in Bangladesh, e.g., mola (*Amblypharyngodon mola*) and chanda (*Parambassis ranga*), have very high vitamin A contents and retinol activity equivalents (RAE) of >2500 and 1500 µg per 100 g raw edible parts, respectively. While concerns over the nutrient supply provided by fishery resources are well-founded, few attempts have been made to investigate the nutrient composition of a wide range of fish and shellfish species and their contribution to nutrient availability at a national level. Estimating food availability in a region and the subsequent macro- and micro-nutrient supplies remain an important part of ensuring food and nutrition security. Moreover, such information can also be used to build a global nutrient supply database [15,16,17].

As the global population continues to grow, so does the demand for seafood for human consumption [18]. Seafood can either be wild-caught or farmed, but fishery yields from wild capture have plateaued in recent years as fisheries stocks are exploited near or greater than their maximum sustainable yields [19,20]. As a result, the aquaculture industry has become the fastest-growing sector of food production in the world, expanding from 0.6 million tonnes (Mt) in 1950 to 80 Mt in 2016 [21,22]. This has triggered questions about how trends in capture fisheries and aquaculture production might respond to different levels of consumption for seafood. What is the optimal scenario (or combination of scenarios) for sustaining the population in the future, if wild capture and aquaculture fisheries were to make an increasing, decreasing, or constant contribution to seafood availability? This question has become increasingly relevant and requires an urgent answer as only 10 years remain of the 2030 Sustainable Development Goals (SDGs) agenda.

This study assesses the contribution of Thai fishery products to the national nutrient supply and examines possible trajectories of seafood availability and levels of seafood consumption. Predictions were based on long-term trends of population growth and wild-caught and farmed fishery yields. We used Thailand as a case study as it is a substantial producer and net exporter of fishery products relative to other countries [21]. Moreover, recently, the Government of Thailand has taken a hard stance against unsustainable fishery practices that have negatively impacted marine ecosystems [23,24]. It is surmised that several regulations and additional measures will lead to changes in the available supply of Thai seafood [23,25,26].

We divide this research into three main sections. First, we present the supply of fishery products in Thailand. Second, we estimate the quantity of seafood-derived nutrients at the national level. Third, we explored the possibility of predicting seafood availability and consumption levels, based on a range of simulated scenarios.

## 2. Data collection and Methods

### 2.1. Data Collection

Secondary ecological data were collected from a broad range of publications (Table 1). The data on wild capture and aquaculture production were obtained from the fisheries statistical yearbooks that have been published by Thailand’s Department of Fisheries (DoF). We used a time series of catches from 1995 to 2015. Subsequently, we followed a study by Nesbitt et al. (2010) [27] to identify the common (vernacular) name, scientific name, genus and/or family of fish and shellfish. Furthermore, we used the databases of Fishbase (http://www.fishbase.org, accessed on 14 January 2021) and the International Union for Conservation of Nature (IUCN) Red List of Threatened Species (http://www.iucnredlist.org/about, accessed on 14 January 2021) to identify the name, scientific name, and family of all the species mentioned in Thailand’s list of fishery products.

### 2.2. Estimation of National Nutrient Supply from Fishery Products

The nutrient content of harvested fish and shellfish were mostly sourced from the ASEAN food composition tables published by the Institute of Nutrition, Mahidol University (2014) [29], and the Thai food composition tables published by the Nutrition Division, Department of Health (2001). Where food composition data were not available, best-matching values were obtained from the nutrient composition tables of other countries, including India [32], Japan [31], the Philippines [30], Malaysia [33], and Vietnam [34]. Additionally, we used the nutrient database for fish and shellfish products from the United States Department of Agriculture (USDA) (2019) [37] and FAO (2016) [35], as well as published documents and scientific articles, such as Fellows and Hampton (1992) [38], Siong et al. (1987) [39], and Tacon and Metian (2013) [40] (Appendix A).

To estimate the edible raw portions of fish and shellfish species, we used data from multiple sources, because the food composition database in Thailand was not able to provide an edible portion of fishery products. We converted the total yield of fishery products to edible weights, as nutrient availability from fishery products is often listed as nutrient content per hundred grams (i.e., density) of edible portion. For species where the data on nutrient content of fish and shellfish at the species level were not available, we used the data of other species belonging to the same genus and/or family following the guidelines of FAO (2016) [35]. In cases where data of the family were also absent, we used the average of the general group that the species belonged to, e.g., shrimp, fish, or crab (Figure 1). All data (i.e., landings, edible portions, and nutrition profiles) were entered into an Excel spreadsheet to assess the national nutrient availability. The descriptive results were presented as mean ± standard deviation (SD).

### 2.3. Scenario Analysis

We carried out a scenario analysis to investigate how future per capita seafood consumption trajectories might be met by wild-caught fisheries and aquaculture production by 2030, given that Thailand aims to fulfil ‘zero hunger’-SDG 2, while ensuring improved ‘life below water’-SDG 14. Hence, our analysis focused mainly on nutritional aspects of Thai seafood consumption and production while economic factors, such as the effects of price formations in domestic and international markets on country-level supply and demand, are beyond the scope of this manuscript. Figure 2 summarizes our scenario analyses, which aimed to estimate the differences between consumption and availability of seafood in Thailand. For all scenarios, we applied the United Nations projections for population growth in Thailand [41], following the article by Merino et al. (2012) [42].

We established five potential scenarios for per capita seafood consumption, including i. maintaining current seafood consumption, ii. healthy and balanced consumption, iii. healthy diets from sustainable food systems, iv. higher consumption, and v. lower consumption. Specifically, the second and third scenarios were based on the eating guide designed for Thai people by the Department of Health (2001) [43] and Willett et al. (2019) [3]; respectively, while the last two scenarios were calculated based on the per capita seafood intake of 187 countries reported in the Global Dietary Database (GDD) (2019) [44] (Appendix A). Note that GDD data suggests that per capita seafood intake was relatively independent from the income level of the countries, as the mean intakes were similar between countries at different income levels (Appendix A).

We proposed four scenarios for predicting the supply of seafood, including i. maintaining current seafood availability from the 2015 baseline, ii. decreasing wild capture fisheries, iii. expansion of aquaculture production, and iv. decreasing wild capture production compensated by an increase in aquaculture production. Our second and third scenarios were based on the proposed plans by the Government of Thailand to avoid overfishing in the Gulf of Thailand, while the last scenario combined both measures concurrently. Also noteworthy is that the domestic seafood supply was calculated following the formula of Smith et al. (2016) [17], in which fish meal production and exported fishery products were deducted from the sum of capture and aquaculture production yields and imported fishery products in Thailand. Thereafter, this value was then divided by the total Thai population to obtain the seafood supply per capita. The formula used is as follows:

Domestic supply of seafood (kg/year) = yield of capture production (kg/year) + yield of aquaculture (kg/year) − quantity used in fish meal production (kg/year) − quantity of exported fishery products (kg/year) + quantity of imported fishery products (kg/year)
(1)


#### 2.3.1. Seafood Destined for Human Consumption

Scenario D1: Maintaining current seafood consumption: Based on the GDD, the average seafood intake in Thailand was 32 g of seafood per day (g/d), or 12 kg/y.

Scenario D2: Healthy and balanced consumption: In this scenario, seafood intake is based on the recommendation by the Nutrition Flag, a healthy eating guide designed for Thai people by the Department of Health (2001) [43]. The guide recommends an individual to consume approximately 30 g/d, or 11 kg/y.

Scenario D3: Healthy diets from sustainable food systems: Willett et al. (2019) [3] suggested that healthy diets from sustainable food systems should contain, on average, 28 g/d, or 10 kg/y.

Scenario D4: Higher consumption level: We used the GDD (2019), which reports the per capita seafood intake in 187 countries. For the higher demand of seafood intake, we used the 95th percentile of the dataset, which was 48 g/d, or 18 kg/y.

Scenario D5: Lower consumption level: We assume that the volume of seafood consumed in Thailand will decrease to the lower quartile (the 25th percentile of the dataset) of the GDD (2019) [44], which is 14 g/d, or 5 kg/y.

#### 2.3.2. Availability of Edible Seafood Products

Scenario S1: Maintaining current seafood availability: In this scenario, we assumed that the trends in yields of wild capture fisheries and aquaculture production, the quantity used for fish meal production, and the quantity of fishery exports and imports will remain stable from the 2015 baseline (Appendix A).

Scenario S2: Decreasing capture fisheries production: Recent assessments of Thailand’s fish stocks estimated that the fishing effort for demersal fish in 2015 exceeded the level which would produce maximum sustainable yield by 33% in the Gulf of Thailand and 5% in the Andaman Sea. Meanwhile, the fishing effort of pelagic fish exceeded the optimum level by 27% in the Gulf of Thailand and 17% in the Andaman Sea [45]. In order to achieve the maximum sustainable yield, we assume that the yield of capture fisheries production will decrease by 13% from approximately 1.5 Mt in 2015 to 1.3 Mt in 2016. This value was calculated based on the targets of Thai fisheries to reduce their fishing efforts from the 2015 baseline. For instance, the government’s plan is to decrease 20% of the fishing effort for demersal fish in the Gulf Thailand. Based on recent stock assessments, the demersal catch in the Gulf of Thailand is 503,276 tons and the fishing effort is 36 million hours, so a reduction of 20% would mean 29 million hours of fishing effort and an optimal catch of 402,621 tons [45]. We considered this for all the categories (e.g., reduction of 5% in the Andaman Sea for demersal fish, reduction of 20% for pelagic fish in the Gulf of Thailand and 10% for pelagic fish in the Andaman Sea). Thereafter, we use these values to calculate the future seafood supply in Thailand.

Scenario S3: Expansion of aquaculture production: According to the current 5-year National Economic and Social Development Plan (2017–2021), the Government of Thailand announced its policy to encourage aquaculture production [46]. Similarly, Gentry et al. (2017, 2019) [1,47] suggested that the development of marine aquaculture at 1% of a country’s suitable ocean area could present an opportunity for increasing aquaculture production. Therefore, we assume that the development of Thailand’s aquaculture production will increase by 1% of the total production yield each year. With similar calculation methods used in Equation (1), we predicted future changes in the availability of seafood for Thai consumption.

Scenario S4: Decreasing capture production compensated by an increase in aquaculture production: This scenario combined the previous two scenarios (Scenario S2 and S3). We consider that the yield of capture fisheries production will decrease by 13% from 2015, while the yield of aquaculture production will increase 1% every year.

It should be noted that the values used in the supply calculations include the weight of non-edible portions, such as bones and shells (Smith et al., 2016). We therefore converted the weights to edible amounts as follows, after estimating the per capita supply of seafood. We took an average of 52% of edible weight of fish, which is the most abundant type of seafood produced in Thailand.

## 3. Results

### 3.1. Supply of Fishery Products in Thailand

From 1995 to 2015, the annual harvest from the fisheries sector in Thailand is officially estimated to be between 2.4 and 4.1 Mt, with an average production yield of 3.4 ± 0.5 Mt [28]. Approximately 71% of the total fisheries production was from wild capture fisheries, while 29% were from aquaculture production. Total wild capture production (including both marine and inland sources) was around 2.4 ± 0.6 Mt on average (range 1.5–3 Mt). Based on the available databases, we report that annual wild capture production went down by 50% over the twenty-year period, from 3 Mt in 1995 to 1.5 Mt in 2015. Meanwhile, Thailand’s aquaculture production rose by 68%, from 0.6 Mt in 1995 to 0.9 Mt in 2015 [28].

Global seafood supply has continuously increased from an average of 15 kg/y in 1995 to 20 kg/y in 2015. Similarly, the average supply of seafood products in Asia has increased from 16 kg/y to 23 kg/y. Although the average seafood supply per capita in Thailand was found to be greater than the average in other Asian countries and globally, there was a decrease from 31 kg/y to 23 kg/y over the same period (Figure 3) [48].

### 3.2. Nutrient Availability from Fishery Products in Thailand

We estimated the national nutrient availability from fishery products (wild capture and aquaculture) in Thailand from 1995 to 2015. Our results indicate that the total harvested fishery products supplied between 19% and 35% of the Thai recommended daily intake (RDI) of protein for Thai people aged six years and above, 4–6% of calcium, 6–11% of iron, and 2–4% of zinc [49].

Among wild capture production, the largest supply of protein and zinc came from the “trash fish” group. The Engraulidae, specifically *Stolephorus* spp., was the largest contributor to the calcium supply. *Sardinella* spp. from the Clupeidae family contributed most to the total iron yield. As for families involved in aquaculture production, the largest supply of protein, calcium, and zinc came from Penaeidae, including *Litopenaeus vannamei*, *Penaeus monodon*, and *Metapenaeus* spp., while Mytilidae, specifically *Perna viridis*, contributed most to the total iron supply.

### 3.3. Future Projections of Seafood Availability and Consumption Levels

Scenarios D1 and D2 represent the current consumption level and recommended consumption, which corresponds to demands of around 820 and 766 thousand tonnes (kt) per year for the entire Thai population, respectively. Scenarios D3, which is related to healthy diets from sustainable food systems, results in a seafood demand of around 715 kt per year. Scenarios D4 and D5 represent the higher and lower bounds for the seafood demand, which are 1237 kt per year and 364 kt per year respectively.

Our predictions of seafood availability estimate that there would be 857 kt/year of seafood under scenario S1. If the yield of capture production were to decrease (Scenario S2), seafood supply would be 755 kt per year. In contrast, if the aquaculture sector continues to expand (Scenario S3), this would lead to 897 kt of seafood per year. Scenario S4 results in an available seafood supply of 796 kt per year.

Finally, we considered different combinations of the levels of seafood consumption and seafood availability scenarios in 2030 (Figure 4). If seafood consumption levels were to increase by 50% (Scenario D4) as compared to current seafood consumption, no supply scenario would provide the required amount of seafood for the Thai population. On the other hand, all seafood availability scenarios would be sufficient to meet sustainable consumption levels (Scenario D3), as well as the lowest possible consumption (Scenario D5). More realistically, if the annual yield of capture fisheries were to decrease, there would be an insufficient supply of seafood for Scenarios D1, D2, and D4. However, if yields of aquaculture production would increase in the future, this might help provide sufficient supply for an increased seafood consumption.

## 4. Discussion

For the scenario analysis, the quantity of seafood for human consumption was predicted to remain relatively stable until 2030, because the Thai population is projected to only increase by 2% within that time period (from 69 million people in 2016 to 70 million in 2030) [50]. Scenarios D1 and D2 result in similar national-level seafood demands per capita, as the current seafood consumption level and the recommended Thai consumption level (i.e., Nutrition Flag) are quite similar. Interestingly, if the Thai per capita seafood intake would increase to the higher possible consumption level, none of the seafood availability scenarios would result in an adequate amount of seafood for the population. In addition, if the annual yields of capture production were to decrease, as is quite likely due to the destruction of oceanic ecosystems and depletion of fish stocks in recent years, there would be insufficient seafood if the Thai people maintain current seafood consumption level or even drop to a recommended intake level for a healthy and balanced diet. An increased aquaculture production may help bridge the deficit and enable increased seafood demand in Thailand. However, it should be noted that the continued expansion of aquaculture production may negatively affect the environment [51,52], hence production would have to be increased in such a way as to minimize environmental degradation. A key challenge of aquaculture is to sustain production within environmental limits. Several efforts have been proposed in the Thai aquaculture industry to promote responsible aquaculture practices. Different trophic levels are integrated during fish farming or aquaculture production is integrated with rice culture and rearing of livestock [53,54]. Also zero-water-exchange systems are implemented [55], to reduce the dependence of the fisheries on the supply of aquaculture feed [56]. These innovative concepts can incorporate both traditional and advanced technologies that are needed to ensure viable long-term solutions for aquaculture production. Several of these solutions do not require high-technology inputs, and are often economically profitable as well [52].

To meet the national demand for animal source foods, production of other major types of meat and animal products (e.g., bovine, pig meat, poultry meat, eggs, and milk) in Thailand has increased gradually from 1995 to 2015, averaging about 4 Mt per year [48]. Overall, the production of pig meat, poultry meat, and animal products (e.g., eggs and milk) increased gradually, while the production of bovine meat decreased slightly during 1995–2015. Although terrestrial animal food production could fulfil a role in provisioning nutritious food, natural disasters and animal disease outbreaks can have a destructive effect on the industry, and thus food supply. For example, in 2004, Thailand’s poultry production was disrupted by disease outbreaks of highly pathogenic avian influenza [57]. Furthermore, several studies indicated that the increasing terrestrial animal-sourced food production can interact with reactive nitrogen as one of the main threats to global climate change [58]. To combat environmental degradation, wide-scale lifestyle changes across populations may be necessary. Dietary changes, such as reducing the consumption of animal-sourced food and adopting vegetarian/vegan options, can lead to reduced greenhouse gas emissions and thus reduced environmental impact [59].

Thai fisheries play an essential role to supply food for the country’s growing population [60,61]. According to the GDD database (2019) [44], the average amount of seafood consumed per capita in Thailand was higher compared to that of other Asian countries such as Nepal, India, and Indonesia. Countries with the highest seafood intake per capita are Japan, the Maldives, and Portugal, with consumption values of 75, 62, and 56 g/d, respectively. In traditional societies, food consumption may depend on what can be acquired and/or grown [62,63]. Geography drives seafood demand, with high seafood consumption in fishing regions.

Thailand’s fisheries have developed substantially over the last decades and contributed to the progress of the SDGs in Thailand via supplying nutritious food and generating economic growth [64]. Based on the DoF’s databases from 1995 to 2015, the estimated total aquaculture production (in both marine and freshwater areas) in Thailand has gradually grown from 0.6 Mt in 1995 to 0.9 Mt in 2015. Meanwhile, marine capture production has decreased by almost half, from 2.8 Mt in 1995 to 1.3 Mt in 2015 during the same period. The drop in marine capture production could be attributed to tightening restrictions by neighbouring countries for access into their exclusive economic zone, and the depletion of resources due to overfishing, environmental degradation, and illegal, unreported and unregulated (IUU) fishing [23,24,65]. Thailand has started to amend fishery laws and deterred illegal fishing activities in order to prevent marine resources from being undermined and to promote a sustainable use of fisheries resources [23,24,65,66]. We acknowledge the need to recognise IUU fishing challenges in Thailand. Strategies to eliminate illegal fishing, reduce fishing effort, and decrease harvest will benefit the development of fisheries in a long term. In a short term, however, aggressive fishery reforms will reduce the amount of marine catch as well as have economic and social repercussions on profits [25]. Likewise, these modifications can have an effect on the demand and supply of nutritional food in the country. Consequently, the decline in the per capita supply of seafood in the Thai population according to the database of FAO (2019a) [36] might be caused by the decline in marine catch. Furthermore, based on the Thailand Fisheries Statistics Yearbook from the DoF, an average of 39% of total fisheries products were utilized for non-food uses such as fish meal.

In view of study limitations, data on seafood species diversity and fish and shellfish composition are useful to estimate the nutritional contribution of fishery products. As we used nutrient composition data from different databases, the data slightly differed depending on factors such as catch season, stage of the life cycle, amount and quality of feed, and other environmental conditions [67,68,69]. Nevertheless, these data can help estimate the nutrient supply and investigate food biodiversity at national-level [68]. Additionally, there have only been a few scientific studies on the nutrient composition of different fish species, even though fish and shellfish provide a significant source of macro- and micro-nutrients. We propose that future work should examine information regarding the nutrient composition of additional species involved in fishery products. In future, these nutritional evaluations might help guide the country’s policies and plans for improving and monitoring population-level nutrient status and safeguarding food security. Furthermore, a considerable seafood loss and waste can have an impact on the national seafood supply. Thailand utilised low value fish, which is often discarded in many other countries, to produce fish meal used as feed for aquaculture or poultry farms, and fish sauce or Budu sauce for human consumption [25,70]. Therefore, the rate of fish discarding is expected to be low [23]. We propose strategies to reduce fish loss and waste involved in consumption patterns. For example, several studies have indicated that the edible portions of fish and cooking methods could affect the actual nutrient content of fish [10]. Due to the lack of data, we were unable to consider the traditional utilisation of fish in diets in the households in our scenarios. Hence, it is recommended that future studies should examine how the traditional utilization of fish in diets can have an impact on the nutrient availability and health of local populations. In addition, other factors, such as trends in the quantity of exported and imported fishery products and the quantity used for fish meal production over time are important when estimating the national supply of seafood. The lack data on these aspects is an important limitation of this study.

Furthermore, several recent studies have indicated the nutritional potential of diverse marine food sources, such as seaweed [71,72,73] and jellyfish [74,75], as they are often a rich source of protein, minerals, and vitamins [71,73,74]. In 2018, farmed aquatic algae or seaweed production was 32.4 Mt of wild-collected and cultivated aquatic algae combined, whereas jellyfish (e.g., *Rhopilema* spp. and *Stomolophus mele*) catches were estimated to be approximately 0.3 Mt [72]. Unfortunately, the national fisheries statistical yearbooks of Thailand do not explicitly report the volume of seaweed production, which current small-scale seaweed farming practices. FAO (2018c) [71] indicated that Thailand imports edible seaweed such as dried wakame, nori, and agar strips and powder for direct human consumption. At present, the Republic of Korea and China are the principal producers of dried edible seaweed.

Moreover, our scenario analysis is limited with regard to key economic factors, such as the price of seafood which is known to influence the national supply. Other factors, which include labour costs, oil prices, and transportation costs, can affect seafood prices [66]. For example, an increase in diesel prices could lead to higher costs in fishing activities [65]. Fishermen would have a lower net benefit, and consequently overall marine capture production would decrease. In addition, the shifts in seafood prices can be affected by the interplay between the supply and demand for seafood products [75,76]. The preference of consumers may also affect the future price of seafood [75]. These factors may reflect the important impacts on national seafood supply. It is also important to acknowledge that scenarios S3 and S4, which consider the potential expansion of Thai aquaculture production, have excluded an increase in domestic fish meal import, which is however likely to be necessary. ’Fishing the feed’ is a growing issue in the sector, as aquaculture often requires a huge amount of wild fish, particularly trash fish, to feed farmed fish and shellfish [77,78]. As such, the sustainability of Thai fisheries might pose a great burden of foreign fish stocks. To address the issue, alternative sources of protein, such as algae meal, wheat gluten, corn gluten, and insects, could in future replace or reduce the use of fishmeal and fish oil in aquafeed production [21,79].

Our results pointed out that distinct taxonomic groups contain different densities of each nutrient. For example, a high concentration of calcium is found in the Engraulidae family (>160 mg per 100 g of raw fish). Small fish species are a significant source of calcium content, as they can often be consumed whole with bones, heads, and viscera through certain processing methods (e.g., drying and deep-frying), as in the case of *Sardinella* spp. The data presented also support studies by Roos et al. (2007) [10], Bogard et al. (2015) [67], Kawarazuka and Béné (2011) [12], and Thilsted (2012) [13], which focused on the nutrient content of certain small fish species to improve nutritional status in developing countries. In addition to being nutritious, small fish are also typically inexpensive; for instance, the average value of anchovies (*Stolephorus* spp. and *Encrasicholina* spp.) in Thailand was 0.3 ± 0.1 US$/kg during 1995–2015 [28]. Marinda et al. (2018) [14] suggested that various species of small fish come from capture fisheries and are crucial to preserve the health of oceans through environmental integration, monitoring and management strategies. In this context, one can consider the use of integrated socio-environmental models to link and balance ecosystem composition and functioning with the diverse functions and the SDG’s based on quantitative evidence [80].

## 5. Conclusions

We estimated the contribution of fishery products to Thailand’s national nutrient availability. We conclude that fishery products have the potential to substantially contribute to the food and nutrition security of Thailand. Since the wide species diversity of harvested fishery products provide different nutrient yields, fish-based nutrition strategies should leverage the distinct types of fish and shellfish species available within an area. Thereafter, we examined several future projections of seafood availability and consumption levels, in which we based the availability and consumption scenarios on population growth and fishery production trends, respectively. We analysed five levels of seafood consumption, while the changes in capture fisheries and aquaculture production were used to predict the seafood availability. Our findings indicate that if the levels of seafood destined for human consumption were to increase drastically up to the level of countries with the highest intake globally, there would be an insufficient quantity of fishery products for the Thai population. We also report that if capture production were to decrease, as is quite likely due to the destruction of oceanic ecosystems and the depletion of fish stocks in recent years, there would not be insufficient supply to accommodate a recommended healthy diet, much less the current Thai diet. Although an increase in aquaculture production can bridge this gap, we must consider the environmental impacts of such an approach. The balance between nutrition and sustainability is a public health challenge that will require concerted efforts from policymakers and the fisheries sector at large.

## Figures and Tables

**Figure 1 foods-10-00880-f001:**
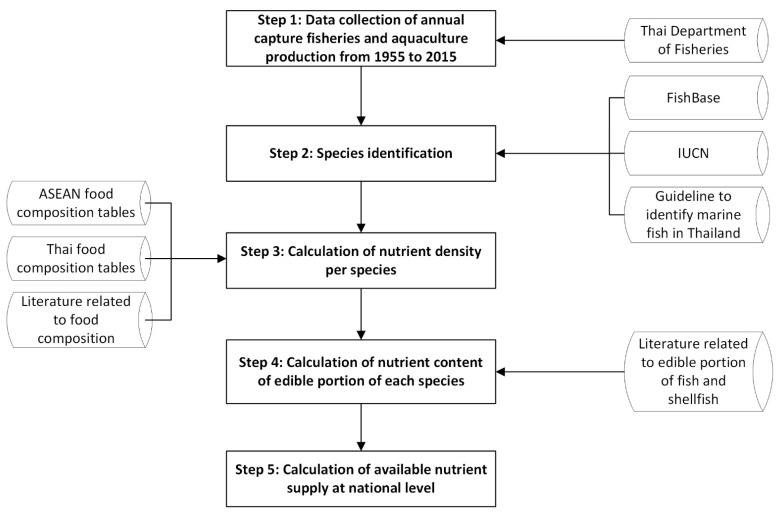
Flow chart of the data collection procedure and estimation of nutrient supplies.

**Figure 2 foods-10-00880-f002:**
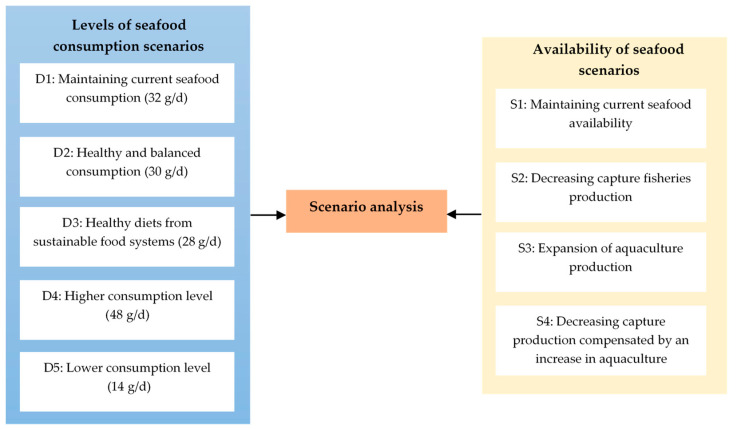
Schematic diagram of the scenario analysis considering different scenarios of Thai seafood availability and consumption.

**Figure 3 foods-10-00880-f003:**
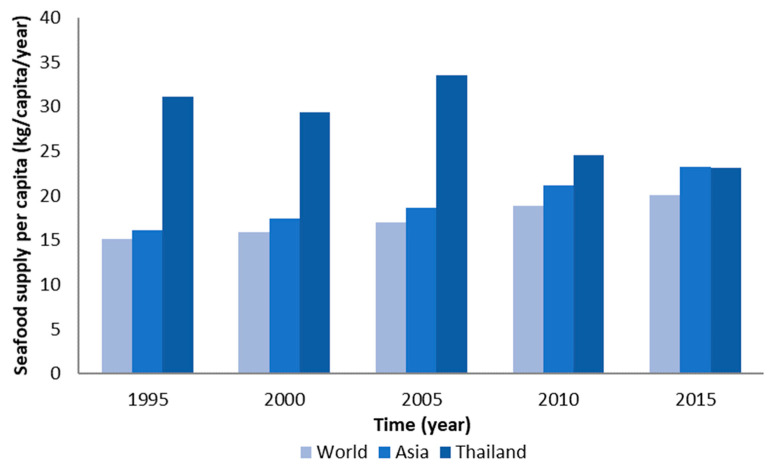
Comparison of the average seafood supply per capita globally, in Asia, and in Thailand, from 1995 to 2015. Source: FAO (2019a) [48].

**Figure 4 foods-10-00880-f004:**
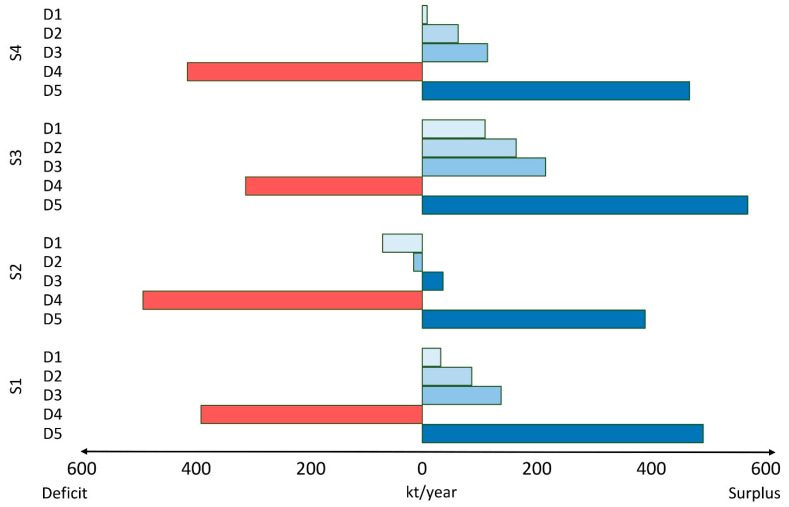
Difference (kt/year) between the levels of Thai seafood availability and consumption scenarios in 2030. D1: Maintaining current seafood consumption (32 g/day), D2: Healthy and balanced seafood consumption (30 g/day), D3: Healthy diets from sustainable food systems (28 g/day), D4: Higher consumption (48 g/day), and D5: Lower consumption (14 g/day). S1: Maintaining current seafood availability, S2: Decreasing capture fisheries (13% of reduction of yield of capture production in 2015), S3: Expansion of aquaculture production (1% of increase of yield of aquaculture), and S4: Combination of S2 and S3. Blue bars represent the surplus of Thai seafood availability and consumption in 2030 while red bars represent the deficit of Thai seafood availability and consumption in 2030.

**Table 1 foods-10-00880-t001:** Data sources. The webpages were all accessed on 14 January 2021.

Database	Description	Link
Thailand’s DoF (1998–2017) [28]	Annual fisheries and aquaculture production from 1995–2015	https://www.fisheries.go.th/strategy-stat/document-public
FAO balance sheet (2019) [7]	Per capita seafood supply The production of meat and animal products (e.g., bovine meat, pig meat, poultry meat, milk, and eggs)	http://www.fao.org/faostat/en/#data/FBS/report
FishBase (2018)	Taxonomic verification of fishery products	http://www.fishbase.org
The IUCN Red List of Threatened Species (2018)	Taxonomic verification of fishery products	http://www.iucnredlist.org/about
Institute of Nutrition, Mahidol University (2014) [29]	ASEAN food composition tables	http://www.inmu.mahidol.ac.th/aseanfoods/composition_data.html
Food and Nutrition Research Institute, Department of Science and Technology (2019) [30]	Philippine food composition tables	https://i.fnri.dost.gov.ph/fct/library/viewfct
Nutrition Division, Department of Health (2001)	Thai food composition tables	http://nutrition.anamai.moph.go.th/ewt_news.php?nid=492
Ministry of Education Culture Sports Science and Technology (2005) [31]	Standard tables of food composition in Japan	http://www.mext.go.jp/en/policy/science_technology/policy/title01/detail01/1374030.htm
Longvah et al. (2017) [32]	Indian food composition tables	http://www.ifct2017.com/frame.php?page=home
Institute for Medical Research (1997) [33]	Malaysian food composition database	http://myfcd.moh.gov.my/
National Institute of Nutrition (2007) [34]	Vietnamese food composition table	http://www.fao.org/fileadmin/templates/food_composition/documents/pdf/VTN_FCT_2007.pdf
FAO (2016) [35]	FAO/INFOODS global food composition database for fish and shellfish, version 1.0 (uFiSh1.0)	http://www.fao.org/infoods/infoods/tables-and-databases/faoinfoods-databases/en/
FAO (1989) [36]	Nutritional value of commercially more important fish species	http://www.fao.org/3/T0219E/T0219E00.htm#TOC
USDA (2019) [37]	Food composition database	https://fdc.nal.usda.gov/index.html

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
