# Peer review of "The Contribution of Thai Fisheries to Sustainable Seafood Consumption: National Trends and Future Projections"

_foods, 2021, doi:10.3390/foods10040880_

Round 1

Reviewer 1 Report

I  am happy with the authoritative and careful way in which the authors dealt  with my comments 

Author Response

The authors would like to thank the reviewer for their insightful remarks and suggestions, which improved our manuscript tremendously.

Reviewer 2 Report

The sustainability problem of aquaculture production expansion is neither solved nor faced because of its pure accountancy conception: in other words, Scenario S3 (1% of increase of yield of aquaculture) implies that either higher fisheries captures or higher fish meal import support the aquaculture production expansion. This is not considered by authors. S4 Scenario (13% reduction of yield capture production combined with S3 Scenario) is realistic only when supported by higher fish meal import. Higher aquaculture production requires high fish meal volumes. It means that in front of the lower domestic fisheries effort, aquaculture expansion can be based only on higher import fish meal volumes. Thi is a hypothesis that implies that Thai fisheries sustainability is loaded on foreign fish stocks' shoulders.

Author Response

Indeed, scenarios S3 and S4, which both consider an expansion of aquaculture production, have excluded an increase in fish meal import. We acknowledge that ‘fishing the feed’ is a growing issue in aquaculture, as it requires huge amounts of wild fish, particularly trash fish, to help feed farmed fish and shellfish industries (Boonyubol & Pramokchutima, 1984; FAO, 2011b). As such, the domestic sustainability of Thai fisheries could pose a burden on other countries, as described by the reviewer. We added this interesting point to our discussion section, to make the reader aware of the potentially negative consequences of aquaculture production expansion. Moreover, we also refer to potential solutions for this unsustainable expansion, i.e. feeding alternative sources of protein, such as algae meal, wheat gluten, corn gluten, and insects, that might replace or reduce the use of fishmeal and fish oil in aquafeed production (Beal et al., 2018; Krogdahl et al., 2010). These points have been added in the revised manuscript as follows.

  • It is also important to acknowledge that scenarios S3 and S4, which consider the potential expansion of Thai aquaculture production, have excluded an increase in domestic fish meal import, which is however likely to be necessary. ’Fishing the feed’ is a growing issue in the sector, as aquaculture often requires huge amount of wild fish, particularly trash fish, to feed farmed fish and shellfish (Boonyubol & Pramokchutima, 1984; FAO, 2011b). As such, the sustainability of Thai fisheries might pose a great burden of foreign fish stocks. To address the issue, alternative sources of protein, such as algae meal, wheat gluten, corn gluten, and insects, could in future replace or reduce the use of fishmeal and fish oil in aquafeed production (Beal et al., 2018; Krogdahl et al., 2010). (Lines 377-384)

Reviewer 3 Report

Very interesting and well-structured study conducted by Sampantamit and co-workers.

This study investigated the supply trend of fishery products and subsequent seafood-derived nutrient availability over the last decade. In addition, it was explored the possibility of predicting seafood availability and consumption levels, including adherence to Thailand’s national food guide and global dietary recommendations for sustainable seafood consumption. The topic is very emergent and the results are of high significance. 

Text is clear and easy to read. I only have minor suggestions :

References are not according to the journal’s instructions.

What is the role of jellyfish and seaweed fisheries in feeding the world and Thai population in the next few years? I think the authors should provide some information about this. I recommend the following literature:

Raposo, A., Coimbra, A., Amaral, L., Gonçalves, A., & Morais, Z. (2018). Eating jellyfish: safety, chemical and sensory properties. Journal of the Science of Food and Agriculture98(10), 3973-3981.

Bonaccorsi, G., Garamella, G., Cavallo, G., & Lorini, C. (2020). A systematic review of risk assessment associated with jellyfish consumption as a potential novel food. Foods9(7), 935.

Cherry, P., O’Hara, C., Magee, P. J., McSorley, E. M., & Allsopp, P. J. (2019). Risks and benefits of consuming edible seaweeds. Nutrition reviews77(5), 307-329.

Palmieri, N., & Forleo, M. B. (2020). The potential of edible seaweed within the western diet. A segmentation of Italian consumers. International Journal of Gastronomy and Food Science20, 100202.

In the Discussion section, it should be provided more studies outside Thailand to compare the results with other results obtained in different regions of the world.

Author Response

Indeed, jellyfish and seaweed fisheries might be a promising solution for increasing human food demand., given that they are a rich source of protein, mineral, and vitamins. Large amount of aquatic algae and seaweed are being cultivated and collected, i.e. 32.4 million tonnes of aquatic algae and  0.3 million tonnes of jellyfish (e.g. Rhopilema spp. and Stomolophus mele) (FAO, 2020). Unfortunately, no data on seaweed production can be found in the national fisheries statistical yearbooks of Thailand, although FAO (2018) indicated that Thailand imports edible seaweed such as dried wakame, nori and agar agar strips and powder for direct consumption. We have added the information, including the suggested references, to our revised manuscript as follows.

  • Furthermore, several recent studies have indicated the nutritional potential of diverse marine food sources, such as seaweed (FAO, 2018c; FAO, 2020b; Palmieri & Forleo, 2020) and jellyfish (Raposo et al., 2018; Bonaccorsi et al., 2020), as they are often a rich source of protein, minerals, and vitamins (FAO, 2018c; Raposo el al., 2018; Palmieri & Forleo, 2020). In 2018, farmed aquatic algae or seaweed production was 32.4 Mt of wild-collected and cultivated aquatic algae combined, whereas jellyfish (e.g. Rhopilema and Stomolophus mele) catches were estimated to be approximately 0.3 Mt (FAO, 2020b). Unfortunately, the national fisheries statistical yearbooks of Thailand do not explicitly report the volume of seaweed production, which might be due to current small-scale seaweed farming practices. FAO (2018c) indicated that Thailand imports edible seaweed such as dried wakame, nori and agar strips and powder for direct human consumption. At present, the Republic of Korea and China are the principal producers of dried edible seaweed. (Lines 357-368)

Round 2

Reviewer 2 Report

The sustainability problem cannot be solved by a sentence. It is much more than a sentence. You would have considered it only by revising the scenarios. The use of alternative fish meal is still experiencing the pioneristic phase.

This manuscript is a resubmission of an earlier submission. The following is a list of the peer review reports and author responses from that submission.

Round 1

Reviewer 1 Report

This paper is an investigation into the problem of maintaining nutritionally healthy diets in Thailand at a time when overfishing and climate change have reduced the catch levels of wild capture fisheries. Although increased production of fish from aquaculture is helping to fill the gap, this comes at a cost of environmental damage. The authors suggest innovative ways of making best use of the seafood that is available in order to maximize the nutritional benefit of eating fish at a time of decreasing supply.

It is extremely well-written with a clear and coherent argument on an important and topical subject; it is impressively researched, based on a sound methodological approach; and it contributes some convincing recommendations for policy-making. My only suggestion to the authors is that they may consider adding a few notes on the following three issues.

First, the authors might mention the steps that could be taken to avoid the waste of fish. For example, a considerable amount of fish is discarded overboard by vessels which engage in high grading. Perhaps Thailand could emulate the EU’s landing obligation policy which forbids discarding in EU waters? Additional measures to reduce waste include improved refrigeration systems on board vessels; more enterprising marketing strategies; and smarter consumption practices. Second, the authors might note the huge amount of illegal, unreported and unregulated (IUU) fishing that takes place in Thai waters, and consider whether or not that widens the gap between the demand and supply of nutritional food in the country. Third, the authors might elaborate more on their observation that aquaculture causes environmental damage. Given that globally, the quantity of fish produced from aquaculture is now greater than the quantity produced from wild capture fisheries, we need to know how to reduce the environmental cost of farming fish.

Reviewer 2 Report

Scenario analysis is carried out without taking into account important economic factors that influence the supply. First of all, future global market fish prices dynamics is ignored. It could influence Thai fish export as well import, thus reflecting important impacts on national supply. Second, Thai GDP and purchasing power is ignored, although it would influence price formation in domestic market whatever demand scenario is proposed. Third, it is not clear if the authors considered the consequences of aquaculture expansion (S3 and S4 scenarios) on fish meal production subtracted to national supply. If not, it is a methodological error, if they did then they would have made explicit the mechanism used in order to consider the fish meal demand. These arguments suggest the adoption of a functional economic model of supply-demand rather than a comparative static balance scenario model.

Round 2

Reviewer 2 Report

The answers in the response letter do not satisfy the need of clarifying the true nature of the problems. In other words: is fish meal production functionally linked to aquaculture production in formula (1)? What is the technical coefficient of the fish meal in the aquaculture production function? The two issues of the formula (1) are simultaneously integrated. It makes Scenario S4 (Decreasing capture production compensated by an increase in aquaculture production) lacking of any sense because of unsustainable growth of aquaculture production without adequate support of capture fisheries production. Scenario S2, too, would have considered the impact of decreasing capture fisheries on aquaculture production. One can say that import fishery products couold compensate the decreasing domestic supply, but I repeat the point: what is the role of prices in the supply dynamics? With reference to this argument, authors acknowledge the limit of ignoring price dynamics in the demand-supply scheme, providing only a weak accountable analytic framework.